# Compassion as a guiding framework for the implementation of digital mental health interventions: An interview study with clients and professionals

Charlotte M. van Lotringen[1]*, Alec Zirnheld[1], Saskia M. Kelders[1,2], Gerben J. Westerhof[1], Matthijs L. Noordzij[1]

**1** Department of Psychology, Health & Technology, University of Twente, Enschede, The Netherlands
**2** Optentia Research Unit, North-West University, VTC, South Africa

\* c.m.vanlotringen@utwente.nl

## Abstract

### Background

Digital mental health interventions are often described in terms of their contribution to cost-effectiveness or innovation. Instead, many clients and professionals in mental healthcare seem to value the human connection highly. To implement technology in ways that align with values held by clients and professionals, a value-based framework for technology use in mental healthcare could be promising. The current study explores whether values of clients and professionals in mental healthcare match a framework of compassion, and whether this framework could be a suitable foundation for the implementation of digital mental health interventions.

### Method

We conducted semi-structured interviews with 5 (former) clients and 15 professionals in mental healthcare. Values of both clients and professionals were analyzed inductively, and deductively linked to a compassion framework. Professionals were asked whether their values were congruent with their organization's approach to technology. We coded their answers as matches and mismatches, and described the themes developed in both categories.

### Results

Values held by clients and professionals showed many connections with the compassion framework. Clients highly valued feeling heard and understood, humanity, and openness from the professional. Professionals highly valued helping people, personalization, and offering transparency. Examples of how technology use could enhance or detract from compassion according to participants were also produced.

**Data availability statement:** While transparency and data sharing are important in research, sharing qualitative data in mental health contexts can raise significant ethical concerns that outweigh the benefits of open access. Even anonymized, the detailed nature of interviews may still make participants—both clients and professionals—identifiable, especially in specific or local settings. Clients often disclosed deeply personal, sensitive experiences, while professionals shared candid reflections on their practices and workplaces. These details risk compromising confidentiality and professional relationships. Therefore, the ethical obligation to protect participants' privacy and psychological safety must take priority over data sharing. Data are available via the corresponding author or via the Ethics Committee (contact via ethicscommittee-hss@utwente.nl) for researchers who meet the criteria for access to confidential data.

**Funding:** This publication is part of the project "designing compassionate technology with high societal readiness levels for mental healthcare" (project number 403.19.229) of the research program Transitions and Behavior, financed by the Dutch Research Council (NWO), Minddistrict BV, and Dimence Groep. CvL, SMK, GJW and MLN received salary from these funds. The funders had no role in study design, data collection and analysis, decision to publish, or preparation of the manuscript.

**Competing interests:** The authors have declared that no competing interests exist.

Professionals experienced a match with their values if they felt that their organizations focused the adoption of technology on the client's autonomy or meeting treatment needs. They experienced a mismatch if they felt that their organizations were more focused on financial benefit or a technology push.

## Conclusion

Compassion seems a promising framework for integrating technology in mental healthcare in value-sensitive ways.

## Introduction

Digital mental health interventions (DMHIs) can play an important role in improving the quality of and access to mental healthcare, as well as meeting the rising demands for care [1]. Digital mental health refers to eMental Health and other technologies that help improve the user's mental health and wellbeing [2], such as online modules and diaries, chatbots, virtual reality, serious games, and online support groups. These DMHIs have the potential to lighten the workload of the professional, to give the client a more active role through self-management, and to offer innovative treatment options [3]. However, their actual use and implementation in mental healthcare is still limited [4].

The way technology is currently being introduced in mental healthcare does not seem to match with the values of stakeholders in this field. While technological innovations are often presented from a viewpoint and motivation of improving organizational efficiency and finances [5–7], clients and professionals prioritize receiving and providing high-quality, personalized care and the human connection [8–10]. This highlights the need to integrate compassion, an essential value in care and defined as recognizing and addressing suffering, as a fundamental element in ensuring healthcare remains human-centered amidst technological advancements [6]. Currently, this value is often missing entirely in discussions about the implementation of technology [6]. The current interview study builds upon previous work studying the connection between compassion and technology [11,12], by examining the values of clients and professionals in mental healthcare and whether compassion could serve as a framework for integrating DMHIs more meaningfully into this field.

### Integrating DMHIs in treatment

The use of DMHIs can be as effective as 'regular' face-to-face treatments [13]. Yet, despite their potential advantages, the acceptance of and adherence to DMHIs remains low [14–16]. Clients and mental health professionals indicate that they highly value a personal connection during treatment [17,18]. In line with this value, research has shown that adding human support to DMHIs can enhance their effectiveness and adherence [19,20]. One way to do this is with blended treatment, where digital components are combined with face-to-face components [21]. Research shows that clients and professionals prefer blended treatment to the standalone use of DMHIs [22,23].

Some advantages that professionals see for blended treatment are that it could make treatment more accessible and motivating for clients, accelerate treatment, and make the therapeutic relationship more intimate and personal [8,24]. They experience that contact in-between the face-to-face sessions can increase the sense of continuity in the treatment, intensify the treatment, and engage clients more [8]. These findings are promising, as a good therapeutic alliance has been associated with better treatment outcomes [25–27], and a higher frequency of contact has been linked to significantly faster recovery in psychotherapy [13,28]. Indeed, a previous narrative review showed that a digital therapeutic relationship is possible and beneficial, and explored how both human support and technological features could contribute to the relationship [29].

While professionals do see the potential of using technology in treatments, it depends on how technology is implemented in practice which benefits or disadvantages are experienced [30]. Professionals report that they lack knowledge and experience on how to integrate DMHIs into their treatments and have to juggle multiple tasks when using DMHIs in treatment [8,16,24]. A clear conceptual basis for how to embed DMHIs into daily practice is lacking, as they are not part of routine care yet [8,16,21,24,31,32]. Moreover, there is the perception that technologies are 'cold' and will be implemented at the expense of 'warm' human care [33]. Earlier studies report professionals' fears that financial efficiency is being prioritized over humanity and patient care in the adoption of blended care [10,34], which can cause the experience of pressure to use DMHIs for financial reasons, in turn leading to distrust [8]. This seems to indicate a mismatch in the approach of management and insurance companies towards the use of DMHIs on the one hand, and the values of mental health professionals on the other hand.

## Importance of values

Values refer to personal or societal judgments of what is valuable and important in life [35], and are closely linked to one's motivations [36]. The intentions and motivations of stakeholders such as clients and professionals are important factors in the success of integrating DMHIs in treatment [37,38]. Therefore, it is paramount to understand the values of professionals and their clients. If the professional's personal values are aligned with organizational values, this has the potential to improve their job satisfaction and reduce burnout [39]. On the contrary, discrepancies between personal and perceived organizational values can impede the ability to be compassionate, and lead to lower job satisfaction and higher burnout [40]. Thus, it is also important to investigate whether the values of professionals match with perceived organizational values around the reasons to use technology.

An essential value in mental health care is compassion [38,41,42]. In short, compassion refers to the recognition of suffering and the wish to relieve this suffering. Based on a review and synthesis of earlier definitions, Strauss et al. [42] proposed that compassion consists of five elements:

"1) Recognizing suffering; 2) Understanding the universality of suffering in human experience; 3) Feeling empathy for the person suffering and connecting with the distress (emotional resonance); 4) Tolerating uncomfortable feelings aroused in response to the suffering person (e.g., distress, anger, fear) so remaining open to and accepting of the person suffering; and 5) Motivation to act/acting to alleviate suffering" [42].

Previous qualitative studies on the needs and experiences of clients in mental healthcare show the importance of feeling actively listened to, to be understood, receive emotional support and non-judgmental acceptance, and help with dealing with problems [43–47]. Therefore, earlier studies advocate for compassion as a central tenet in care from the perspective of the client [43,44]. However, the voices of mental healthcare clients are still lacking in current mental healthcare policy [47].

In contrast with research on the client perspective, qualitative studies on the values of professionals are sparse. However, this research is of interest to understand the influence of therapists' factors on the process and outcome of therapies [48], and therefore also to understand why DMHIs are or are not being adopted by them [7]. One recent study forms an exception, by using interviews with 15 Dutch therapists to explore their career choice motivations [49]. When asked what made these mental health professionals choose their career, participants mentioned (among other reasons) being interested in understanding other people, having an empathetic disposition, and wanting to help and give back. Moreover,

when asked what influences them in daily practice, participants mentioned (among other factors) a sense of equality and respect in establishing connections with their patients, and the need for resilience to deal with work-related distress. These themes match with the five proposed elements of compassion. Nevertheless, among professionals, beliefs also exist that compassion is draining or even not useful [40]. Therefore, it is possible that compassion is not a shared key value among all professionals.

Although prior work, such as Hodges et al. [50], has highlighted the centrality of compassion in healthcare, its potential as a framework to analyze the values of professionals and clients in mental health care and its suitability as a conceptual basis for implementing DMHIs remain underexplored. Similarly, there has been limited research on how the values of mental health professionals align or conflict with their organization's approach to technology. Addressing these topics could help to better understand the implementation process of DMHIs. Thus, the current study explores these themes using semi-structured interviews. More specifically, the current study will look into the following research questions:

1. What are values of clients in mental healthcare, and how do their values and experiences with DMHIs fit in the compassion framework?

2. What are values of professionals in mental healthcare, and how do their values and experiences with DMHIs fit in the compassion framework?

3. How do the values of professionals correspond with their perceptions of how their organizations present technology?

In short, the aim of the current study is to explore whether values of clients and professionals in mental healthcare match a framework of compassion. If compassion emerges as a suitable framework for the values of both clients and professionals, it could provide the conceptual foundation needed to guide technology implementation strategies. This would support the meaningful and effective integration of DMHIs into mental healthcare while maintaining a focus on human connection and care.

## Methodology

In reporting this study, the standards for reporting qualitative research (SRQR) checklist [51] was followed (S1 Table).

### Participants

Participants were 5 mental healthcare clients who were in treatment at the time of the interview or had been in treatment <1 year before the interview and 15 mental healthcare professionals who were actively working (partly) for mental healthcare organizations at the time of the interview. All participants received or gave treatment within the Dutch mental healthcare context. Participants were recruited through convenience sampling. They participated on a voluntary basis, and received no reward. Clients had a mean age of 33.4 (min: 20, max: 60), and were all female (n = 5). Mental healthcare professionals had a mean age of 42.9 (min: 22, max: 57), with the majority being female (n = 12). See Table 1 for detailed participant demographic data.

### Materials

To answer our research questions, we conducted semi-structured interviews [52] via MS Teams or on location. The interviews followed an interview guide. The interview guide for clients (S1 File) contained questions about their experiences and needs in treatment, how their treatment was shaped, the roles of them and their therapist, and if and how any DMHIs were used. The interview guide for mental healthcare professionals (S2 File) contained questions about their work motivations, difficult aspects of their job, their values in work, how they shaped their treatments, if and how they made use of any DMHIs in their treatments, and if their values aligned with how their organization presented the added value of technology. ATLAS.ti (Version 24.1.0 (30570)) was used for qualitative data analysis.

**Table 1. Participant Demographic Data.**

| Characteristic | | N |
|---|---|---|
| **Mental healthcare professionals (N = 15)** | | |
| Age (years) | | |
| | 18–30 | 2 |
| | 31–50 | 9 |
| | 51–60 | 4 |
| Gender | | |
| | Female | 12 |
| | Male | 3 |
| Job Experience (years) | | |
| | <1 | 1 |
| | 1–3 | 4 |
| | 4–9 | 3 |
| | 10–19 | 5 |
| | >20 | 2 |
| Profession | | |
| | Clinical Psychologist | 2 |
| | Clinical Psychologist & Psychotherapist | 1 |
| | Healthcare Psychologist | 4 |
| | Nurse Specialist | 2 |
| | Orthopedagogue | 1 |
| | Psychiatrist | 1 |
| | Psychologist | 3 |
| | Social Psychiatric Nurse | 1 |
| Types of technology mentioned | | |
| | Apps | 3 |
| | Biofeedback | 1 |
| | Chat/SMS/Email | 8 |
| | Digital EMDR (online or with VR) | 2 |
| | Electronic Patient File | 2 |
| | Robots | 1 |
| | Online modules | 15 |
| | Video calling | 11 |
| | VR | 7 |
| | Wearables | 1 |
| Role of technology in treatment | | |
| | Combination of face to face and digital | 8 |
| | Mainly digital | 2 |
| | Mainly face to face | 3 |
| **Mental healthcare clients (N = 5)** | | |
| Age (years) | | |
| | 18–30 | 3 |
| | 31–50 | 1 |
| | 51–60 | 1 |
| Gender | | |
| | Female | 5 |
| | Male | 0 |

*(Continued)*

**Table 1.** (Continued)

| Characteristic | | N |
|---|---|---|
| Type of mental healthcare received | | |
| | Basic | 2 |
| | Specialist | 3 |
| In treatment for (could be a combination) | | |
| | Anxiety | 1 |
| | Autism Spectrum Disorder | 3 |
| | Depression | 1 |
| | Emotion regulation | 1 |
| | Post Traumatic Stress Disorder | 2 |
| | Stress | 1 |
| Types of technology mentioned | | |
| | eHealth modules | 3 |
| | Email | 1 |
| | Care robot | 1 |
| | Mobile apps | 3 |
| | Online or digital EMDR | 2 |
| | Videocalling | 1 |
| | Websites with information | 2 |
| | Wearables | 2 |
| Role of technology in treatment | | |
| | Combination of face-to-face and digital | 4 |
| | Mainly digital | 0 |
| | Mainly face-to-face | 1 |

## Procedure

Ethical approval for this study was granted by the ethics committee of the faculty of Behavioural, Management and Social Sciences of the University of Twente (registration number 221229). All participants signed written informed consent forms before participating. We did not assess participants' capacity to consent, as we had no reason to believe that their capacity to give consent was impaired by any factor, such as illness or disability.

The semi-structured interviews took place between the 25th of October 2022 and the 24th of April 2023. They were conducted online (n = 18) or on location (n = 2) during one-to-one sessions that lasted an average of 51 minutes (std. dev. 13 minutes, min.: 28 minutes, max.: 82 minutes). The sessions were recorded (screen and audio or audio). The interviews were transcribed verbatim using Amberscript's automated service [53] and corrected and anonymized manually. The recordings of all interviews were stored on a secured hard drive.

In case the interviewee was a (former) mental healthcare client, the researcher emphasized that they did not have to answer questions they did not feel comfortable with, that their answers would not influence their treatment trajectory, and that details of personal treatment trajectories would not be discussed.

## Data analysis

The semi-structured interviews were analyzed using two reflexive thematic analyses [54], one on the interviews with clients and one on the interviews with professionals. The first author (CvL) and a student assistant (psychology master student) first familiarized themselves with the interview data, systematically identifying relevant quotes in an inductive

manner. Then, they inductively generated initial codes. CvL and a different psychology master student (AZ) reviewed the codes and sorted these into overarching themes. These codes and themes were discussed until agreement was reached, and then defined and named in several iterative cycles by CvL and AZ. Using the final coding schemes, AZ coded 15 percent of the interview data, with a percentage agreement between CvL and AZ of 90%.

We used different parts of the overall coding schemes to answer the different research questions. To answer the first research question, we used codes and themes from the full interviews with clients. To answer the second research question, we focused on codes related to professionals' answers to the interview questions about what motivates them in their job, what difficult elements are in their job, and what their values are for their work. Additionally, to answer the first and second research question, the codes we developed were linked in a deductive manner to one or more of the five elements of compassion if fitting. This was done by two researchers (CvL and AZ), by comparing the content of the codes to the definitions of the elements of compassion proposed by Strauss et al. [42], with an inter-rater agreement percentage of 92%. In the same way, CvL and AZ matched quotes about the use of technology to compassion elements if fitting. If we did not see a clear link between a code on values or a quote about technology use and one or more of the compassion elements, no compassion element was selected.

To answer the third research question, we used codes related to professionals' answers to the interview questions about how they use technology in their treatments (if at all), how their organisation presents technology, and what they think of that approach. Then we deductively coded the relevant quotes as either a match or a mismatch (between the professional's value(s) and their perception of their organization's technology approach). Within the match and mismatch themes, we then inductively developed codes that described the (mis)match.

### Researcher reflexivity

One researcher conducted the interviews (CvL) and two researchers were mainly involved in the data-analysis (CvL and AZ). CvL is a Dutch woman born in the Netherlands and raised in a non-religious environment. She holds a Master's degree in Psychology and has previous experience with conducting and analyzing semi-structured interviews. She has a strong interest in the mental healthcare context and how mental health professionals and clients experience and shape their treatments. Compassion is a key value to her in personal life as well. While conducting the interviews, she strived for an open and non-prescriptive tone to create space for the stories and perspectives of participants, which led to in-depth and highly personal conversations. Compassionate technology for mental health care is the topic of CvL's PhD research project, supervised by MLN, SMK and GLW. The aim of this project was to investigate how technology in mental healthcare can be integrated into daily practice, where compassion is a fundamental value. This project took place in collaboration with researchers from different disciplines, a mental health care organization, and an eHealth company. AZ is a Luxembourgish-French person born in Luxembourg and raised in a multi-cultural, non-religious environment. He moved to the Netherlands to pursue his studies and holds a Master's degree in Psychology. His Master's internship and thesis were conducted under the guidance of MLN and CVL within the compassionate technology project.

### Results

We will describe the results for each of the three research questions one by one.

**1. What are the values of clients in mental healthcare, and how do their values and experiences with DMHIs fit in the compassion framework?**

From an inductive perspective, we divided client's values into two main themes: professional-focused values and treatment-focused values. We will first briefly discuss these two themes, and then link the values to the compassion framework proposed by Strauss et al. [42]. Table 2 gives a comprehensive overview of all the themes and codes, the number of clients that spoke about a code, and the related element of compassion.

**Table 2. Overview of the Themes and Codes of Client's Values.**

| Theme | Code | N of clients | Element of Compassion |
|---|---|---|---|
| **Professional-Focused Values** | Feeling heard and understood | 5/5 | Recognition of suffering, Empathy |
| | Humanity | 5/5 | Common humanity |
| | Openness/distress tolerance of professional | 5/5 | Distress tolerance |
| | Personal connection and empathy | 5/5 | Empathy |
| | Availability and motivation of professional to help | 3/5 | Acting to alleviate suffering |
| **Treatment-Focused Values** | Support outside of treatment room | 5/5 | Acting to alleviate suffering, Common humanity |
| | Balance of autonomy vs. support | 4/5 | |
| | Personalization of treatment | 3/5 | Recognition of suffering, Acting to alleviate suffering |
| | Gradual phasing out of treatment | 2/5 | |
| | Positive/strength-based approach | 2/5 | |
| | Reliable information | 1/5 | |

## Professional-focused values

Professional-focused values refer to qualities of the mental healthcare professional, or the client-professional relationship. The first professional-focused value that was mentioned by all participants, was feeling heard and understood. Participants talked about how important it was to (finally) feel understood, and to feel taken seriously. The value of humanity was also mentioned by all participants, referring to the experience of equal human-to-human contact. All participants appreciated it when their therapists were open to their emotions, and gave room for difficult feelings and experiences, as is illustrated in the code "openness/distress tolerance". If this was lacking, clients could feel fear of overburdening the professional: "I've also had [therapists] in past treatments- yes, I call them those fragile dolls. That I thought: 'I better not say that today' […], that you have such an unstable person, so to speak" (C5). Furthermore, all participants mentioned a personal connection and empathy, and described appreciating genuine, compassionate reactions from a mental healthcare professional.

## Treatment-focused values

Treatment-focused values referred to important qualities of the treatment. Almost all participants spoke about the importance of a balance between autonomy and being supported, describing shared decision-making, receiving more directive support when needed, and being able to remain autonomous. Three participants mentioned valuing the personalization of their treatment, rather than feeling like a simple protocol was being followed: "They [the therapist] really looked at […] how I am as a person. She really made the treatment very personal" (C4). Furthermore, all participants appreciated different kinds of support outside of the treatment room, such as being in touch with other experience experts, the involvement of family members, or using email or apps to receive support.

## Compassion as a framework for clients' values and technology use

See Fig 1 for a visual map of the codes and their links to compassion elements. Participants often spoke about elements of compassion as proposed by Strauss et al. [42] in their own words when describing experiences with mental health professionals, for example: "They [mental healthcare professionals] really made sure that I felt comfortable to tell my story, and were very helpful" (C3). We could also link some of the elements of compassion to clients' descriptions of different types of technologies used in their treatment.

**Recognizing suffering.** The compassion element of recognizing suffering was linked to two values held by clients: feeling heard and understood, and personalization of treatment. One participant described it as follows: "My therapist

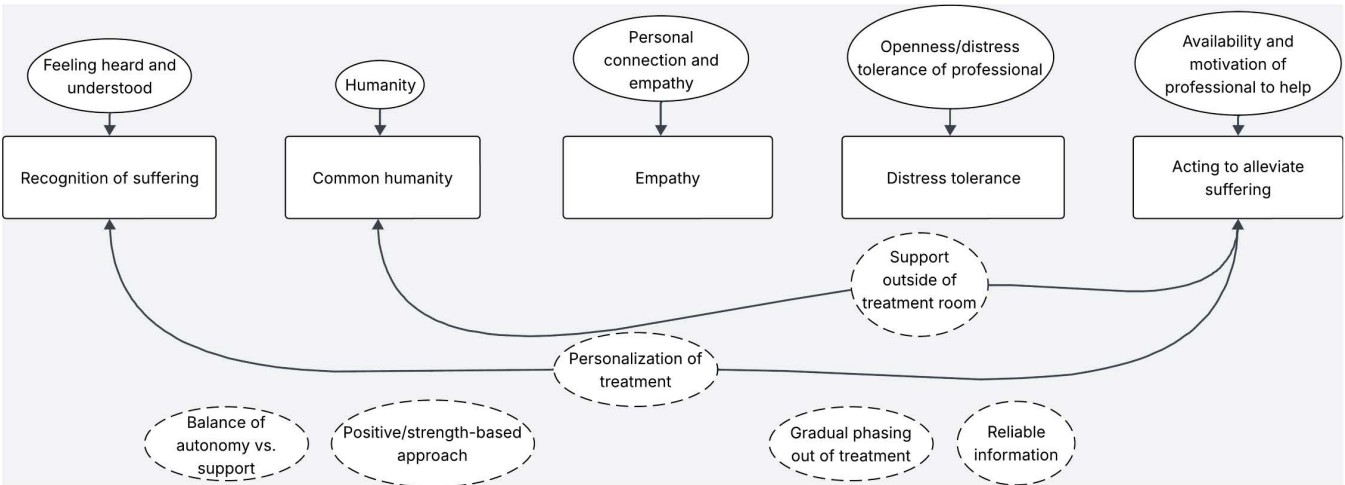

**Fig 1. Visual map of clients' codes and their links to elements of compassion.** The squares contain elements of compassion. Circles with uninterrupted outlines represent codes in the theme 'Professional-focused values'. Circles with dashed outlines represent codes in the theme 'Treatment-focused values'. Arrows represent links with compassion elements.

always keeps their finger on the pulse" (C1). Technology was mentioned several times in relation to recognizing suffering, from being able to hide suffering more easily during video calling, to receiving a message from their therapist after having done something difficult to ask how it went, being able to structure thoughts and feelings in an email to the therapist, and receiving feedback on online modules. One participant gave a smartwatch as an example to be more aware of themselves. To make the most of the use of these technologies to support a professional's awareness, participants stressed the importance of a therapist's responsiveness to their messages or other digital inputs. Finally, one participant described seeing the potential of video calling to stay in touch with the therapist while making the most of the available treatment hours.

**Common humanity.** The element of common humanity was linked to the values humanity and receiving support outside of the treatment room. Humanity was mostly discussed in relation to the mental healthcare professional, to describe the human connection with the therapist: "That is the nice thing about my psychiatrist. There is just human to human contact on an equal footing" (C5). This participant also nuanced this 'equal footing' further: "You [a therapist] should not downplay the suffering, because I really hate that" (C5). Clients also described feeling accepted and not stigmatized by their therapist. One participant described finding common humanity in a group session with other experience experts, which was guided by a therapist. The role of technology was mentioned in only one instance, where a participant reported that YouTube videos of other experience experts helped them to realize that others might experience similar situations.

**Empathy.** The element of empathy, or emotional resonance, was linked to the values of feeling heard and understood, as well as personal connection and empathy. A client describes: "I really got- in my opinion- a sincere response, and that she [the therapist] also genuinely found it annoying for me that I can't do certain things that I really like to do" (C4). Clients described noticing that a mental healthcare professional really felt with them. Technology was not mentioned in relation to empathy.

**Distress tolerance.** The element of distress tolerance was linked to the value openness/distress tolerance of the professional: "I could really tell them [mental health professional] everything […] and they always responded normally, calmly, talked about whatever it was" (C1). Most participants described feeling like they had the space to share anything, and they would receive a professional response and not be brushed off. The perceived absence of a professional's

distress tolerance was mentioned as a barrier in treatment. Technology was not mentioned in relation to distress tolerance.

**Acting to alleviate suffering.** The final element of compassion, acting to alleviate suffering, was connected to the values of the availability and motivation of the professional to help, personalization of treatment, and support outside of the treatment room. One participant described it as follows: "I really learned a lot from [treatment] […] I became very aware and you really receive tools" (C2). This element of compassion was often linked to types of technology. One participant described having mixed feelings towards online modules, seeing the potential: "In the therapy session you have to have your story ready right away so it's nice to be able to type it all out, I think" (C3), but also experiencing them as a stressor and an obligation similar to homework. Another participant described not having found online modules helpful, since they had already read a lot of psychoeducation themselves and did not feel motivated to complete the module. Two participants mentioned positive experiences with online or digital EMDR treatment, which helped to release fearful thoughts. Finally, two participants spoke about trying out a mobile phone app to track their mood, but abandoning it due to feeling like it was an obligation and not finding it helpful enough. Regarding the use of technology, participants described appreciating advice from their professional on which DMHIs to use and valued shared-decision making.

**2. What are the values of professionals in mental healthcare, and how do their values and experiences with DMHIs fit in the compassion framework?**

From an inductive perspective, the professionals' values could be divided into three themes: Treatment-Focused, Client-Focused, and Professional-Focused values. Again, we will first briefly discuss these three themes, and then link the values and mentions of technology use to the compassion framework proposed by Strauss et al. [42]. Table 3 gives a comprehensive overview of all the themes and codes, the number of professionals that spoke about a code, and the related element of compassion.

### Treatment-focused values

Treatment-focused values refer to values around providing treatment to the client and ultimately improving their mental health, as well as the treatment manner and quality. Almost all participants mentioned 'helping people' as a value. One mental healthcare stated that "[the work] makes sense if I notice that the [client] benefits from it" (P8). The importance of personalization rather than tight protocols was also stressed by almost all participants: "Of course we have a whole bunch of guidelines that we have to follow officially. [...] I know those, so I always have them in the back of my mind. But in the end I always try to make it tailor-made" (P11). Nine participants emphasized high quality care as their first priority and the reason they chose their profession, and were bothered when they felt that this priority was undermined, for example by health insurance parties, government policies, and bureaucracy. Eight mental health professionals emphasized the importance of providing transparent treatment to patients, including an open communication towards the client. Moreover, participants valued promoting the client's independence. Professionals spoke about their aim to provide the necessary tools to their clients to take control of their treatment.

### Client-focused values

This theme refers to values around the client's individuality or the client-professional relationship. Six participants spoke about valuing the personal connection highly: "The effect of the treatment starts with the contact. That bridge that you can build, the understanding or encouragement or the comfort, the explanation, that connection, that is just the icing on the cake" (P15). Furthermore, six participants spoke about the value of respect in their relationship with clients. Some professionals expressed the importance of recognizing the equality between them and the client and showing this in their actions. Professionals also mentioned appreciating the variety in their work, in terms of the diversity of clients and tasks: "If it's monotonous it gets boring, then I feel like a conveyor belt employee very quickly" (P11). Other values that were mentioned by some participants were trust, responsibility towards their clients, and being able to give full attention to clients.

**Table 3. Overview of the Themes and Codes of Mental Healthcare Professionals' Values.**

| Theme | Code | N of professionals | Element of Compassion |
|---|---|---|---|
| **Treatment-Focused Values** | Helping people | 13/15 | Acting to alleviate suffering |
| | Personalization over protocol | 12/15 | Acting to alleviate suffering |
| | Quality of care priority | 9/15 | Acting to alleviate suffering |
| | Transparency | 8/15 | Common humanity |
| | Active and Autonomous Client | 6/15 | Acting to alleviate suffering |
| | Efficient (short term) treatment | 1/15 | |
| | Being Available | 1/15 | |
| **Client-Focused Values** | Feeling Connection | 6/15 | Empathy |
| | Respect | 6/15 | Common humanity |
| | Equality | 5/15 | Common humanity |
| | Variety | 3/15 | |
| | Trust | 2/15 | |
| | Responsibility | 2/15 | Distress tolerance |
| | Full attention | 1/15 | Recognizing suffering, distress tolerance |
| **Professional-Focused Values** | Curiosity | 8/15 | Recognizing suffering |
| | Challenge/Puzzle | 8/15 | Recognizing suffering, acting to alleviate suffering |
| | Remaining balanced | 6/15 | Distress tolerance |
| | Eagerness to learn | 6/15 | |
| | Teamwork | 5/15 | |
| | Work/Life Balance | 2/15 | |
| | Pioneering | 2/15 | |
| | Work Freedom | 1/15 | |
| | Enjoyment | 1/15 | |
| | Competence | 1/15 | Acting to alleviate suffering |

## Professional-focused values

Finally, professional-focused values refer to the professionals' personal interests and their standing within the organization. Many participants mentioned finding curiosity important in their work. Related to that, twelve professionals mentioned enjoying the analytical challenge of their job, often comparing it to a 'puzzle': "I also find it a fun intellectual challenge, that puzzle every time again of: how is this person put together? How do I get this person moving?" (P5). Six participants spoke about the importance of remaining in balance emotionally, despite a high workload and personal stress. Furthermore, six participants spoke about being eager to keep learning in their job. The role of colleagues was also mentioned by some participants, who appreciated their connection with colleagues and loyalty to their team. Other professional-focused values that were mentioned less were work/life balance, pioneering, work freedom, enjoyment of the work, and competence.

## Compassion as a framework for professionals' values and technology use

See Fig 2 for a visual map of the codes and their links to compassion elements. Many of the values that participants spoke about could be linked to elements of compassion as proposed by Strauss et al. [42], often quite literally: "I think that it is a kind of basic motivation that I want to help people and have a kind of sympathy for people who suffer" (P5). Furthermore, we connected elements of compassion to how participants spoke about their daily work, both if they worked mostly face-to-face in the treatment room and if they worked (fully) digitally.

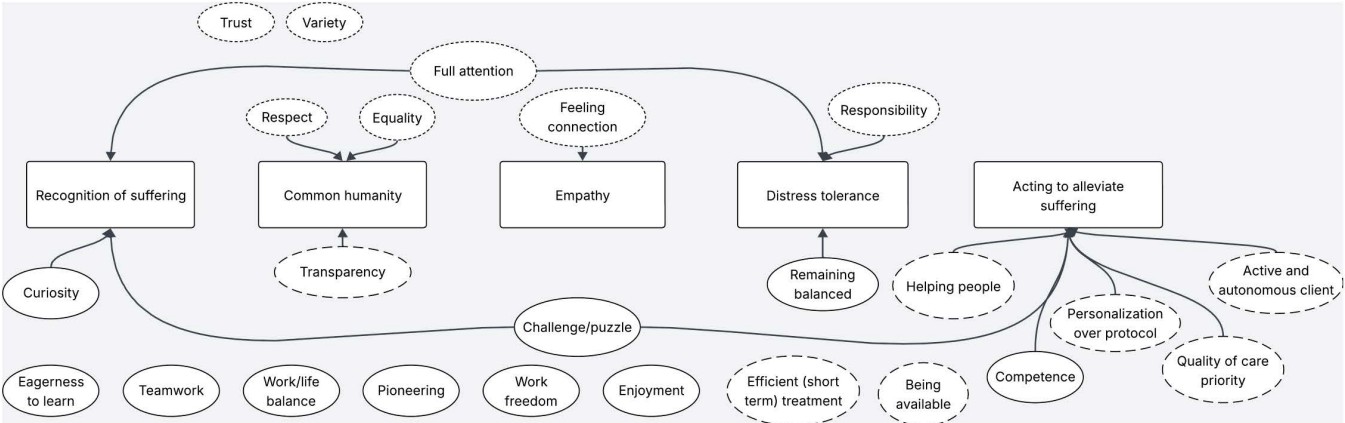

**Fig 2. Visual map of professionals' codes and their links to elements of compassion.** The squares contain elements of compassion. Circles with dashed outlines represent codes in the theme 'Treatment-focused values'. Circles with dotted outlines represent codes in the theme 'Client-focused values'. Circles with uninterrupted outlines represent codes in the theme 'Professional-focused values'. Arrows represent links with compassion elements.

**Recognizing suffering.** The first element of compassion refers to the awareness of suffering: recognizing when and how another person is suffering. The recognition of suffering could be linked to the values of curiosity, being challenged, and giving full attention to the client: "Another important value is that I really try to see the client for who he or she is and what is bothering her, and I can only do that if I am there with my full attention" (P13). In these quotes, the role of technology was prevalent. Participants referred to email, chat, modules, video calling, online diaries, apps, and online questionnaires to help them understand what their clients are going through. Technology could also be used to compare 'objective' and subjective data: "We are currently testing an app about ecological momentary assessment, because I have a client who […] always has a tense feeling on her chest and she thinks that it is constantly present. I would actually like to know [what happens] if I send such a message at random moments and she then has to decide whether the feeling is there or not" (P3). However, one participant questioned whether suffering could be recognized as easily via digital tools as with contact in the same room: "I think that with contact via screens, [the client] can better keep what's really happening hidden" (P9).

**Common humanity.** Common humanity refers to understanding that suffering is universal in the human experience, and therefore not something to judge others for. We identified several values mentioned by professionals that relate to common humanity: equality, transparency, and respect. On the one hand, participants mentioned the importance of a certain equality in the relationship with their clients: "A non-judgmental attitude. Everyone experiences their own things. Who am I to have an opinion about that? And very much together. So not that hierarchical thing of: I am a therapist and I supposedly know how it should be done" (P1). While equality was seen as important, some participants stressed that this did not mean that the relationship with their clients was fully equal: "The other person feels bad and you should not think that this is still an equal relationship. Someone does not come to the clinic for fun. The other person is vulnerable" (P6). A few times, professionals spoke about how technology supports this value: "What I also like about that online module is that it often contains stories of experiences that are then recorded with actors, but still. That is something that you as a therapist can of course use less. You can empathize, but you can't always imagine what it is like" (P1).

**Empathy.** Empathy refers to empathizing and connecting with another person's distress: "How valuable it is that you can really connect with the other. Really, well, as good as a person can, because ultimately we can never completely put ourselves in someone else's shoes, but that you try to connect as closely as possible with the other" (P13). Empathy was linked to the value of feeling connection. The role of technology in this element of compassion was experienced differently

by participants, sometimes as a barrier and sometimes as a facilitator. For example, a nurse specialist described having to use video calling during the COVID-19 pandemic: "With the lockdown, we also had to sit at home and video call. [...] I couldn't feel anything. A lot of your work is also: Look what's vibrating through that room and that doesn't work with video" (P6). However, a psychologist working fully digitally spoke about using short, considerate messages ('nudges') to connect with their clients in between therapy sessions.

**Distress tolerance.**  This element of compassion refers to the ability to tolerate your own uncomfortable feelings in response to a suffering person, so that you can remain open and accepting to the person suffering. When talking about the difficult sides of their job, this element came up (among others): "You hear intense stories from people. That affects you too. You have to think again about: What have I heard? What does it do to me? What does that mean? Yes, you name it, so you literally need some space in that to be ready yourself again" (P10). This element of compassion could be linked to the values of responsibility, giving a client full attention, and remaining balanced. To cope with difficult feelings, participants described supervision and intervision with colleagues, as well as keeping a personal diary. The role of technology was not mentioned in relation to distress tolerance.

**Acting to alleviate suffering.**  The element of compassion that we linked to quotes most often was acting to alleviate suffering: "What motivates me is when treatments work, so when you see that someone who is very stuck in either coping or in complaints, becomes freer and looser from that" (P2). Acting to alleviate suffering could be linked to several values: helping people, personalization, prioritizing high quality care, stimulating clients to be active and autonomous, being challenged to find the best ways to alleviate suffering, and competence. Technology was often mentioned in relation to this element. Mostly, participants spoke about how technologies offered new ways to provide more effective treatment or low threshold contact, such as the use of 3MDR (EMDR combined with virtual reality), video calling, chat, ecological momentary assessment with an app, emailing and texting. A few times, technology was seen as a barrier for alleviating suffering, for example in the case of a nurse specialist treating people with depression during the COVID-19 pandemic: "What I did find is that the technology was in my way here. All that video calling, I thought: yes, really, I can't do anything with it, I'm not helping anyone with it. They have to go through the front door, they have to get moving. That's also what the depression protocol says" (P6).

**3. How do the values of professionals in mental healthcare correspond with their perceptions of how their organizations present technology?**

Overall, 31 quotations about a (mis)match with the perceived organisational approach were extracted. Of these, roughly half indicated a match and the other half indicated a mismatch. Several 'match' themes and 'mismatch' themes were developed, which are shown in Table 4.

### Matches

Three participants spoke about how the way their organization presented technology could support their value of promoting an active and autonomous client. Furthermore, three participants mentioned that their organization's use of technology was in line with (some of) their values, as it could help meet the needs of both clients and professionals better. Another three participants explained that their organization's approach to technology matched with their vision of using technology as a treatment tool, offering extra options to support them and their clients. Two professionals spoke about their organization's approach matching with their values of transparency, for example if clients had access to everything that happens in their treatment. Additionally, one participant said that their organization's approach matched with their value of enjoyment for trying out new things and technologies.

### Mismatches

According to four of the interviewed professionals, their organizations' presentation of technology as bringing financial benefits did not match with their values. In general, participants spoke about the tension between finances and providing

**Table 4. Matches and Mismatches between Professionals' Values and the Perceived Organisational Approach to Technology.**

| Theme | Code | N of professionals |
|---|---|---|
| Matches | Active & autonomous client | 3/15 |
| | Meeting needs | 3/15 |
| | Technology as a treatment tool | 3/15 |
| | Improve transparency | 2/15 |
| | Enjoyment | 1/15 |
| | Normalizing | 1/15 |
| | Improve treatment | 1/15 |
| Mismatches | Financial benefit | 4/15 |
| | Technology push | 3/15 |
| | Loss of human connection | 3/15 |
| | Lack of personalization | 1/15 |
| | Lack of shared vision | 1/15 |

the best care, fearing that a large focus on finances could take the upper hand. One participant explained: "I think that it is often all mixed up. The financial interests and the substantive interests. And if you want to do something new with your employees, such as digital possibilities, you have to try to motivate them, from within […] Starting a conversation about how that fits within your vision and your working method. Well, -that conversation-, I don't think we've even really had that in the team yet" (P14).

Related to this, three participants experienced the way their organization presented technology as a technology push, that was not in line with their values: "I mainly feel the pressure from the organization that it [the use of eHealth] has to be done, because - yes, now I'm putting it bluntly - because that is a requirement of the health insurer, or something. […] Not because we necessarily think it is such a good idea in terms of content, so to speak" (P14). One professional adds that management pushes to use technology, such as video calling, to replace the current face-to-face treatment and explains: "In practice, you also find out that [video calling only] is not what clients want. That is not always the wish that colleagues have. […] A bit of job satisfaction can also be lost if you want to impose [the use of eHealth] too much" (P1).

Another mismatch between the organization's approach to technology and the professionals' values was the loss of human connection when using technology, which was mentioned by three participants. Two participants indicated missing concrete support from their organization on how to convey their values, such as trust and respect, to their clients via DMHIs. Furthermore, one participant spoke about finding it harder to personalize eHealth modules for their clients, which mismatched with their value of authenticity. This participant also described that involving very specific target groups in the development of the modules can help make them more authentic.

## General discussion

The current study made use of semi-structured interviews to explore the values of clients and professionals in mental healthcare, and to connect them to a framework of compassion where possible. Building on previous work that integrated multiple perspectives on compassion and technology [11,12], this approach provides a meaningful foundation for exploring how values in mental healthcare align with or diverge from this framework. Furthermore, the current study investigated congruence and tension between the values of mental healthcare professionals and their perceptions of their organization's approach to technology, giving further insight into the interplay between human-centered care and technological innovation.

In short, we found that clients' values in mental healthcare closely align with the five elements of the compassion framework [42], emphasizing feeling heard and understood, humanity, and openness of the professional. Similarly,

professionals' values reflect compassionate care, including helping people, personalization, and maintaining a human connection. Outside of the compassion framework, professionals also spoke about values related to their personal work enjoyment, such as variety in their work and eagerness to learn. Furthermore, professionals identified both congruence and tension with how their organizations present technology, with alignment seen in fostering client autonomy but mismatches emerging when financial motives or a 'technology-push' overshadow human-centered care. These findings will be discussed in light of relevant related research and their implications for technology implementation in mental healthcare.

First of all, we found that client's values related to the mental healthcare professional closely resembled all five elements of compassion. Indeed, previous studies that did not specifically study a framework of compassion have shown that clients do not only express a need for clinical expertise in professionals, but even more so an empathic attitude: a professional who shows a caring and kind attitude and listens to their clients [43–46]. This finding matches with the aspects of a professional's listening ear, humanity, openness and empathy that all participants in the current study spoke about. Furthermore, among clients' treatment-focused values, almost all clients spoke about the right balance between their autonomy and the amount of support they wished to receive, and how this could differ along their treatment. Again, this is very similar to the finding by Laugharne et al. [46] about the need for a dynamic balance of power between the client and professional, depending on illness severity and the moment in treatment.

Regarding the professionals' values for their work in mental healthcare, we found treatment-focused, client-focused, and professional-focused values. There is overlap between the values currently identified and earlier research that focused on job motivations and needs of mental health professionals. Earlier research also found the motivation to help others, be a responsible person, have genuine, authentic and close interactions with others, but also work enjoyment, variety, intellectual stimulation, self-growth, and professional autonomy [49,55,56]. However, earlier work has not examined current work values, or linked these to a framework of compassion, nor connected them to experiences with technology use. Our findings show that a compassion framework does not only relate to many of the values held by professionals, but can also be linked to ways of using digital mental health interventions (DMHIs), making it suitable to guide the value-sensitive implementation of DMHIs.

Furthermore, the current study shows both matches and mismatches between the values of mental healthcare professionals and how they perceive their organization's approach to technology. Our findings paint a nuanced picture: some aspects of organizational technology policies are already perceived as matching with values of professionals, and both clients and professionals mentioned the use of a wide variety of DMHIs and experiencing their potential. For example, professionals saw matches with their values if their organizations presented technology as a way to stimulate an active and autonomous client, to meet the needs of clients and professionals, or as an extra tool to offer more treatment options. These matches are similar to previous research on potential benefits of the adoption of mobile health technology identified by mental health professionals [34].

However, we also found instances of mismatches that could complicate the integration of DMHIs in treatment. Mismatches were found if professionals felt like their organizations advocated the use of technology for financial benefits, as a technology-push to be an innovative organization, or if the organizational approach seemed to undermine the human-to-human connection. The experience that technology is promoted for financial benefit has been reported before and can lead to distrust from the professional in their organization [8,10,34]. From these results, it becomes clear that professionals are not opposed to the use of technology, but that the way they perceive technology is approached by their organizations is of importance. Some participants in the current study mentioned explicitly that they wish to be involved in meaningful conversations about their organization's motivations for adopting technology, to be sure that its implementation would match with their and their clients' needs.

Based on these findings, it is advisable to take the values of clients and mental healthcare professionals explicitly into account in the implementation of DMHIs. While attention for compassion in digital approaches has been growing in the

field of healthcare in general, for example in Canadian healthcare policy [6], this has not been the case in the context of mental healthcare policies. Taking values into account in the design and implementation of DMHIs could help to align professionals' personal values with organizational values, leading to less risk of burnout, higher job satisfaction, and even a higher ability of professionals to deliver compassionate care [39,57]. The framework of compassion elements based on the proposed definition by Strauss et al. [42] that we used covers many of the values expressed by clients and professionals, and could be a helpful start when discussing the integration of technologies in practice. However, this framework does not match with all of the values produced here. Other values, such as the right balance between autonomy and support valued by clients and the variety in their work valued by professionals, are also important to take into consideration.

Additionally, we found that many professionals in the current study expressed the wish for more clear guidelines for *how* to use *which* technology *when* in their daily practice. The majority expressed a strong preference for guidelines and recommendations rather than strict protocols, to allow for personalization and work freedom. This is in line with earlier work discussing the preference of most clinicians to combine different theories and treatment types in an eclectic manner, and highly appreciating professional independence and autonomy [58]. The way that mental healthcare organizations are often organized can lead to tension between bureaucratic processes and the professional's autonomy [59]. On the contrary, if professionals experience a sense of control over their work and can have input in organizational decisions, this contributes to job satisfaction and better quality of care [59]. Therefore, providing professionals with support while respecting their professional expertise and autonomy is crucial.

### Strengths and limitations

To the best of our knowledge, this is the first study to explore in an in-depth, qualitative manner the values of mental healthcare clients and professionals and link them to elements of compassion and technology use. This work is of importance as current policy in the field lacks attention for client's voices [47], and to improve our understanding of professional behaviour and values for a better implementation of DMHIs [7].

While we were able to conduct interviews with a diverse sample of 15 mental healthcare professionals, recruiting (former) clients proved more difficult, leading to 5 interviews with clients. However, the needs of clients have been a topic of previous research and therefore we mainly aimed to validate these earlier findings. Moreover, many of the themes discussed by participating clients recurred in multiple interviews, showing that a certain saturation had already been reached. Still, it is important to note that the clients who participated all received treatment face-to-face from a professional instead of fully digitally. Moreover, the majority of participants were female. Regarding professionals, this is consistent with the gender distribution in healthcare professions, where women represent the vast majority of practitioners [60]. While compassion is a multi-component process involving the recognition, emotional resonance, and alleviation of suffering, studies have shown that women tend to report stronger affective compassion reactions [61] and that different brain areas were activated between men and women in response to suffering [62]. It is possible that female clients and professionals experience compassion differently from their male counterparts.

### Future research

Future studies could involve managers and policy-makers from mental healthcare organizations. While their more practical motives to promote technology have been studied [7], it would be helpful to explore their values around the implementation of digital mental health interventions. That would make it possible to examine where similarities and discrepancies lie between their values and those of clients and professionals, in order to work towards a shared vision. In addition, it could bring to light whether the way that professionals perceive their organization's approach to technology matches with the actual values behind this approach.

Furthermore, more research is needed to clarify how the implementation of technology in mental healthcare could be shaped by the values of stakeholders. For example, value-based co-design methods [63,64] could be used to involve stakeholders in the development of guidelines or implementation materials for the use of digital mental health interventions. To evaluate how such an approach influences actual experiences of compassion when working with digital interventions, the recently developed Compassionate Technology Scale for Professionals (CTS-P) could be used [12]. Such studies could potentially contribute to clear and compassionate ways of working with technology in daily practice.

## Conclusions

The current study shows that many of the values held by clients and professionals in mental healthcare, respectively surrounding their treatment or work, are closely linked to the different elements of compassion. While some values of mental health professionals are in line with their view of their organization's approach to technology, others are not. Based on these results, it seems promising to reconsider organizational motives for the implementation of digital mental health interventions within a compassionate framework, to evaluate how policies can be more in line with values of clients and professionals. However, to do so realistically, open communication about an organization's financial health and the need to meet a rising demand for care is also essential. Additionally, the co-development of clear guidelines around technology use could support professionals in incorporating technology in ways that reinforce their and their client's values. This could ultimately contribute to compassion-driven mental healthcare, that is supported by the possibilities technology offers.

## Supporting information

**S1 Table. SRQR checklist.**
(DOCX)

**S1 File. Interview guide for semi-structured interviews with (former) clients in mental healthcare.**
(DOCX)

**S2 File. Interview guide for semi-structured interviews with professionals in mental healthcare.**
(DOCX)

## Acknowledgments

We would like to thank the participants for their contributions.

## Author contributions

**Conceptualization:** Charlotte van Lotringen, Saskia M. Kelders, Gerben J. Westerhof, Matthijs L. Noordzij.

**Data curation:** Charlotte van Lotringen.

**Formal analysis:** Charlotte van Lotringen, Alec Zirnheld.

**Funding acquisition:** Matthijs L. Noordzij.

**Methodology:** Charlotte van Lotringen, Saskia M. Kelders, Gerben J. Westerhof, Matthijs L. Noordzij.

**Project administration:** Matthijs L. Noordzij.

**Supervision:** Saskia M. Kelders, Gerben J. Westerhof, Matthijs L. Noordzij.

**Validation:** Alec Zirnheld.

**Writing – original draft:** Charlotte van Lotringen.

**Writing – review & editing:** Charlotte van Lotringen, Alec Zirnheld, Saskia M. Kelders, Gerben J. Westerhof, Matthijs L. Noordzij.

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
