## [Decision Letter · Decision Letter 0]

27 Jun 2025

Dear Dr. van Lotringen,

Thank you for submitting your manuscript to PLOS ONE. After careful consideration, we feel that it has merit but does not fully meet PLOS ONE’s publication criteria as it currently stands. Therefore, we invite you to submit a revised version of the manuscript that addresses the points raised during the review process. 

Please thoroughly adress the review comments (attached at the bottom of this email), and revise accordingly.

We look forward to receiving your revised manuscript.

Kind regards,

Weifeng Han, PhD

Academic Editor

PLOS ONE

Journal Requirements:

2. Please describe in your methods section how capacity to provide consent was determined for the participants in this study. Please also state whether your ethics committee or IRB approved this consent procedure. If you did not assess capacity to consent please briefly outline why this was not necessary in this case.

“This publication is part of the project “designing compassionate technology with high societal readiness levels for mental healthcare” (project number 403.19.229) of the research program Transitions and Behavior, financed by the Dutch Research Council (NWO), Minddistrict BV, and Dimence Groep. CvL, SMK, GJW and MLN received salary from these funds. The funders had no role in study design, data collection and analysis, decision to publish, or preparation of the manuscript”

Reviewers' comments:

Reviewer's Responses to Questions

**Comments to the Author**

1. Is the manuscript technically sound, and do the data support the conclusions?

Reviewer #1: Partly

2. Has the statistical analysis been performed appropriately and rigorously?

Reviewer #1: N/A

3. Have the authors made all data underlying the findings in their manuscript fully available?

Reviewer #1: No

4. Is the manuscript presented in an intelligible fashion and written in standard English?

Reviewer #1: Yes

Reviewer #1: Thanks for sending this interesting paper for review. I found the topic to be very current, the questions well-formed and articulated and the findings interesting. My comments are designed to improve the ms further and appear in the order they arose while reading the ms.

1. The main limitation of the paper in my view is that only 5 clients contributed to the data (in contrast to 15 professionals), but this is acknowledged with reasons. There is also a big question in my mind about the suitability of using frequency counts to the extent that they appear in the ms for this type of methodology (See comment 12 below)

2. I thought that the ms could benefit from mention of the recent work on the digital therapeutic alliance concept (e.g. Tremain H, McEnery C, Fletcher K, Murray G. The Therapeutic Alliance in Digital Mental Health Interventions for Serious Mental Illnesses: Narrative Review JMIR Ment Health 2020;7(8):e17204 doi: 10.2196/17204) . Most obviously from line 70-90 of the introduction where therapeutic alliance is mentioned and perhaps again in the discussion as context for and implications of the findings on the compassion framework. How might the authors’ findings inform the development of digital technologies themselves to create or enhance the sense of compassion from within the digital intervention itself?

3. Furthermore, raising awareness of the work on digital therapeutic alliance could help to shift attitudes towards DMHIs particularly within less convinced clinicians.

4. Line 168 – error in referencing.

5. I would recommend that the authors read, reference and incorporate the recommendations from the RTARG paper and corresponding checklist - which now represents best practice reporting in thematic analysis. [Braun V, Clarke V. Supporting best practice in reflexive thematic analysis reporting in Palliative Medicine: A review of published research and introduction to the Reflexive Thematic Analysis Reporting Guidelines (RTARG). Palliative Medicine. 2024;38(6):608-616. doi:10.1177/02692163241234800]

6. Another limitation on the sample was that it contained only female clients mostly female professionals. Although this is mentioned in the limitations I don’t think this is done appropriately. It is inevitable that a different sample would produce different perspectives – as the aim of qualitative research is to give one set of perspectives, not to produce generalisable or replicable findings. Instead there should be some reflection on how this largely female perspective might contextualise the findings. What might be the unique links or connections between being female and the findings ?

7. Please add information on how ‘professional’ was defined (e.g with a clinical practice qualification, or a set amount of training or just experience or other?)

8. Please justify why and how EMDR counted as a digital technology? This therapy can be entirely non digital.

9. Sentence line 215-6 is unclear and poorly worded (‘If a code….’) - please expand on what is precisely is meant.

10. The Researcher reflexivity (better header than ‘researcher characteristics’) statement should go further: detail how the researcher's background, experiences, and perspectives may have influenced the research process and findings. This includes acknowledging potential biases, assumptions, and values, and explaining how these were considered and managed throughout the study. The statement should also address the impact of the research on the researcher, including any emotional or personal responses. See the RTARG reference above.

11. Table what? L247. ‘2’ missing

12. Using frequency counts in relation to themes has been long criticised – not just by RTARG (where using frequency counts as a justification for themes presented is listed as not appropriate). Throughout the paper frequency counts are used to support the interpretation and findings. This is arguably, bringing a quantitative lens to a qualitative method and is largely shunned by qualitative experts. They are listed in tables, in text and in theme headings. As an essential minimum I’d suggest removing them from the theme titles; if they are retained in narrative text and tables then a strong justification should be provided – but the best solution is probably to remove them entirely (or demote to supplementary files.

13. While the results are interesting they are also very long and text heavy. To aid getting the main messages across I would suggest a thematic map or other diagram or visual representation. I found the current tables a bit too granular and too many to get an overall picture of the findings – which is where a visual could help.

**Do you want your identity to be public for this peer review?** For information about this choice, including consent withdrawal, please see our Privacy Policy

Reviewer #1: No

---

## [Author Response · Author response to Decision Letter 1]

31 Jul 2025

Dear editor,

Thank you for the clear overview of comments. Below we respond to each comment:

1. We have ensured that the manuscript meets PLOS ONE's style requirements.

2. We have now described in our methods section that capacity to consent was not assessed and why.

3. We have clarified our Funding Statement, by adding that no external funding was received for this study, and included the amended Funding Statement in our cover letter.

4. While transparency and data sharing are important in research, sharing qualitative data in mental health contexts can raise significant ethical concerns that outweigh the benefits of open access. Even anonymized, the detailed nature of interviews may still make participants—both clients and professionals—identifiable, especially in specific or local settings. Clients often disclosed deeply personal, sensitive experiences, while professionals may share candid reflections on their practices and workplaces. These details risk compromising confidentiality and professional relationships. Therefore, the ethical obligation to protect participants' privacy and psychological safety must take priority over data sharing. Data are available via the corresponding author or via the Ethics Committee (contact via ethicscommittee-hss@utwente.nl) for researchers who meet the criteria for access to confidential data.

5. We have included captions for our Supporting Information files at the and of our manuscript.

Kind regards,

Charlotte van Lotringen, also on behalf of co-authors

Dear reviewer,

Thank you for your invested time, valuable review comments and suggestions. The review comments were very clear and in-depth, showing that close attention has been paid to the manuscript. We feel that thanks to your attention, the manuscript has been improved further.

Below, please find the original comments and our point-by-point responses.

1. The main limitation of the paper in my view is that only 5 clients contributed to the data (in contrast to 15 professionals), but this is acknowledged with reasons. There is also a big question in my mind about the suitability of using frequency counts to the extent that they appear in the ms for this type of methodology (See comment 12 below)

Response: As can be read in response to comment 12, we can understand your hesitancy about using frequency counts and have removed them from all tables and headings.

2. I thought that the ms could benefit from mention of the recent work on the digital therapeutic alliance concept (e.g. Tremain H, McEnery C, Fletcher K, Murray G. The Therapeutic Alliance in Digital Mental Health Interventions for Serious Mental Illnesses: Narrative Review JMIR Ment Health 2020;7(8):e17204 doi: 10.2196/17204) . Most obviously from line 70-90 of the introduction where therapeutic alliance is mentioned and perhaps again in the discussion as context for and implications of the findings on the compassion framework. How might the authors’ findings inform the development of digital technologies themselves to create or enhance the sense of compassion from within the digital intervention itself? Response: Thank you for this helpful suggestion, this paper is relevant and an interesting read. We now refer to this paper in the introduction, line 83: “Indeed, a previous narrative review showed that a digital therapeutic relationship is possible and beneficial, and explored how both human support and technological features could contribute to the relationship (29).”

3. Furthermore, raising awareness of the work on digital therapeutic alliance could help to shift attitudes towards DMHIs particularly within less convinced clinicians.

Response: We appreciate your suggestion. However, after careful consideration we decided not to incorporate this point in the discussion to preserve the current structure and flow of the discussion. We appreciate your insight nonetheless and hope the rationale for our decision is understandable.

4. Line 168 – error in referencing.

Response: Thank you for pointing this out. We have removed this error (line 173).

5. I would recommend that the authors read, reference and incorporate the recommendations from the RTARG paper and corresponding checklist - which now represents best practice reporting in thematic analysis. [Braun V, Clarke V. Supporting best practice in reflexive thematic analysis reporting in Palliative Medicine: A review of published research and introduction to the Reflexive Thematic Analysis Reporting Guidelines (RTARG). Palliative Medicine. 2024;38(6):608-616. doi:10.1177/02692163241234800]

Response: Thank you for sharing this helpful source. We have familiarized ourselves with it, and now refer to it in the manuscript (line 203) and adjusted the manuscript to fit better with the current best practices in thematic analysis, such as by removing frequencies and adapting headers and wording.

6. Another limitation on the sample was that it contained only female clients mostly female professionals. Although this is mentioned in the limitations I don’t think this is done appropriately. It is inevitable that a different sample would produce different perspectives – as the aim of qualitative research is to give one set of perspectives, not to produce generalisable or replicable findings. Instead there should be some reflection on how this largely female perspective might contextualise the findings. What might be the unique links or connections between being female and the findings ?

Response: Thank you for this helpful observation. We have adapted the description of limitations according to your suggestions, describing potential gender differences in compassion (discussion, line 650: “Moreover, the majority of participants were female. Regarding professionals, this is consistent with the gender distribution in healthcare professions, where women represent the vast majority of practitioners (60). While compassion is a multi-component process involving the recognition, emotional resonance, and alleviation of suffering, studies have shown that women tend to report stronger affective compassion reactions (61) and that different brain areas were activated between men and women in response to suffering (62). It is possible that female clients and professionals experience compassion differently from their male counterparts.”)

7. Please add information on how ‘professional’ was defined (e.g with a clinical practice qualification, or a set amount of training or just experience or other?)

Response: That is a valid point. Professionals were defined as individuals working (partly) for a mental healthcare organization and providing mental health treatments to clients. We have clarified this in line 165: “Participants were 5 mental healthcare clients who were in treatment at the time of the interview or had been in treatment <1 year before the interview and 15 mental healthcare professionals who were actively working (partly) for mental healthcare organizations at the time of the interview. All participants received or gave treatment within the Dutch mental healthcare context.”

8. Please justify why and how EMDR counted as a digital technology? This therapy can be entirely non digital.

Response: You are right, thank you for pointing this out. We have now specified that we are referring to digital ways of applying EMDR, as participants described online or digital EMDR or EMDR with virtual reality (VR).

9. Sentence line 215-6 is unclear and poorly worded (‘If a code….’) - please expand on what is precisely is meant.

Response: Indeed, this sentence could have been phrased more clearly, and we have changed it to: “If we did not see a clear link between a code on values or a quote about technology use and one or more of the compassion elements, no compassion element was selected” (line 222). This line is intended to explain what we did if we did not see a link between codes or quotes and the compassion elements.

10. The Researcher reflexivity (better header than ‘researcher characteristics’) statement should go further: detail how the researcher's background, experiences, and perspectives may have influenced the research process and findings. This includes acknowledging potential biases, assumptions, and values, and explaining how these were considered and managed throughout the study. The statement should also address the impact of the research on the researcher, including any emotional or personal responses. See the RTARG reference above.

Response: We have now renamed this statement ‘Researcher reflexivity’, and extended this section further based on your helpful suggestions (line 232).

11. Table what? L247. ‘2’ missing.

Response: Thank you for your attention to detail. It seems that something went wrong with the internal referencing in the document. This line should refer to Table 2. We have now corrected this (see line 260).

12. Using frequency counts in relation to themes has been long criticised – not just by RTARG (where using frequency counts as a justification for themes presented is listed as not appropriate). Throughout the paper frequency counts are used to support the interpretation and findings. This is arguably, bringing a quantitative lens to a qualitative method and is largely shunned by qualitative experts. They are listed in tables, in text and in theme headings. As an essential minimum I’d suggest removing them from the theme titles; if they are retained in narrative text and tables then a strong justification should be provided – but the best solution is probably to remove them entirely (or demote to supplementary files. Response: Thank you for your valuable suggestion. After familiarizing ourselves with the RTARG, we have removed frequency counts from theme titles and tables.

13. While the results are interesting they are also very long and text heavy. To aid getting the main messages across I would suggest a thematic map or other diagram or visual representation. I found the current tables a bit too granular and too many to get an overall picture of the findings – which is where a visual could help.

Response: We agree that a visual representation would be helpful. Therefore, following your advice, we have added visual maps of the codes for both clients and professionals, see Fig 1. and Fig 2.

Kind regards,

Charlotte van Lotringen, also on behalf of co-authors

---

## [Decision Letter · Decision Letter 1]

7 Sep 2025

Compassion as a guiding framework for the implementation of digital mental health interventions: an interview study with clients and professionals.

PONE-D-25-08553R1

Dear Dr. van Lotringen,

We’re pleased to inform you that your manuscript has been judged scientifically suitable for publication and will be formally accepted for publication once it meets all outstanding technical requirements.

Kind regards,

Weifeng Han, PhD

Academic Editor

PLOS ONE

Additional Editor Comments (optional):

Reviewer #1:

Reviewers' comments:

Reviewer's Responses to Questions

**Comments to the Author**

Reviewer #1: All comments have been addressed

2. Is the manuscript technically sound, and do the data support the conclusions?

Reviewer #1: Yes

3. Has the statistical analysis been performed appropriately and rigorously?

Reviewer #1: N/A

4. Have the authors made all data underlying the findings in their manuscript fully available?

Reviewer #1: No

5. Is the manuscript presented in an intelligible fashion and written in standard English?

Reviewer #1: Yes

Reviewer #1: The authors have been very responsive to all the comments . Thank you. Data availability statement is well justified

**Do you want your identity to be public for this peer review?** For information about this choice, including consent withdrawal, please see our Privacy Policy

Reviewer #1: **Yes: ** Professor Jenny Yiend

---

## [Editor Report · Acceptance letter]

PONE-D-25-08553R1

PLOS ONE

Dear Dr. van Lotringen,

I'm pleased to inform you that your manuscript has been deemed suitable for publication in PLOS ONE. Congratulations! Your manuscript is now being handed over to our production team.

Kind regards,

on behalf of

Dr. Weifeng Han

Academic Editor

PLOS ONE